# Approach to Pancreatic Head Mass in the Background of Chronic Pancreatitis

**DOI:** 10.3390/diagnostics13101797

**Published:** 2023-05-19

**Authors:** Sidharth Harindranath, Sridhar Sundaram

**Affiliations:** 1Department of Gastroenterology, Seth GS Medical College and King Edward Memorial Hospital, Mumbai 400012, India; 2Department of Digestive Diseases and Clinical Nutrition, Tata Memorial Hospital, Homi Bhabha National Institute, Mumbai 400012, India

**Keywords:** pancreatic cancer, chronic pancreatitis, biopsy, endoscopic ultrasound

## Abstract

Chronic pancreatitis (CP) is a known risk factor for pancreatic cancer. CP may present with an inflammatory mass, and differentiation from pancreatic cancer is often difficult. Clinical suspicion of malignancy dictates a need for further evaluation for underlying pancreatic cancer. Imaging modalities remain the mainstay of evaluation for a mass in background CP; however, they have their shortcomings. Endoscopic ultrasound (EUS) has become the go-to investigation. Adjunct modalities such as contrast-harmonic EUS and EUS elastography, as well as EUS-guided sampling using newer-generation needles are useful in differentiating inflammatory from malignant masses in the pancreas. Paraduodenal pancreatitis and autoimmune pancreatitis often masquerade as pancreatic cancer. In this narrative review, we discuss the various modalities used to differentiate inflammatory from malignant masses of the pancreas.

## 1. Introduction

Pancreatic cancer, which has one of the lowest 5-year survival rates of all GI malignancies, is a major contributor to cancer-related death, accounting for approximately 11% of all cancer-related deaths. Smoking, which doubles the chance of pancreatic cancer, long-term adult-onset diabetes, which boosts the risk by 50% to 100%, and other genetic illnesses, which affect 5–10% of cases, are all known risk factors. Less than one-third of all cases of pancreatic cancer can be attributed to one or more of these risk factors. A known risk factor for the occurrence of pancreatic cancer is chronic pancreatitis (CP). Patients with CP have an over 16-fold increased chance of developing pancreatic ductal adenocarcinoma (PDAC), especially within the first 2 years after the diagnosis of CP, with an overall incidence of up to 5% [1].

Imaging may show inflammatory pancreatic abnormalities that resemble pancreatic ductal adenocarcinoma, making a precise preoperative diagnosis difficult, and resulting in unnecessary surgery. Chronic pancreatitis, localised autoimmune pancreatitis, and para-duodenal pancreatitis, sometimes known as “groove” pancreatitis, are examples of inflammatory diseases that might resemble malignant masses [2]. Elderly patients frequently experience mass-forming chronic pancreatitis, with the head of the pancreas being the most prevalent site. However, pancreatic ductal adenocarcinoma also manifests in the same age group and has comparable clinical features, making it difficult to differentiate between the two entities [3].

## 2. Epidemiology and Demographic Comparisons between CP and PDAC

The incidence of CP is 5–12/100,000 people worldwide each year. This closely resembles the 6–12/100,000/year global incidence of PDAC. PDAC affects 4–7% of persons with CP, with patients typically being around 70 years of age, and usually diagnosed within the first 20 years of diagnosis of CP [4]. Case-control, cohort, and record linkage studies carried out over the past few decades provide evidence for this connection. It is recognised that some of the risk factors for chronic pancreatitis also lead to pancreatic cancer. For instance, it is well-known that smoking increases the chance of both diseases, whereas alcohol increases the risk of pancreatitis but not pancreatic cancer. Hereditary and tropical pancreatitis carry the highest relative and lifelong risks of malignancy; hence, the presence of either of these diagnoses should urge rigorous investigations to detect malignancy at an earlier stage.

## 3. Typical Form of PDAC—A Primer

PDAC typically manifests in the sixth or seventh decade of life as dull, painful upper abdominal pain accompanied by nausea, loss of appetite, and gradual weight loss. At the time of presentation, more than 90% of patients with pancreatic cancer experience pain or jaundice [1]. Due to their proximity to the bile duct and ampulla, lesions in the head of the pancreas may present symptoms earlier than lesions in the body and tail [1]. Jaundice is usually not a symptom of distal pancreatic cancer until it has spread, and the condition may be painless until that point. Back or abdominal pain may or may not be present in the early stages of the disease and should not be interpreted as an indication that the tumour is unresectable. However, it has been identified as an independent predictor of mortality [4]. Steatorrhea is a less common symptom. Apart from a palpable gallbladder, physical findings of early illness are seldom evident. Cachexia, ascites, left supraclavicular adenopathy, palpable abdominal mass or liver, and migrating thrombophlebitis are manifestations that point to an advanced disease.

## 4. Clinical Suspicion of Malignancy in CP—When to Suspect?

Slight deviations in the disease’s natural course and/or sudden changes in symptomatology should raise the possibility of a malignant aetiology and warrant thorough investigation. Older age, chronic jaundice, worsening abdominal pain, gastric outlet obstruction, significant weight loss, and a CA 19-9 level greater than 300 IU/mL are clinical characteristics and biochemical markers that suggest a malignant mass in the head of the pancreas [5]. There are some caveats to be kept in mind when using CA 19-9 as a diagnostic marker for pancreatic cancer. As a diagnostic marker, it is neither sensitive nor specific. Additionally, patients with negative Lewis antigen blood type are non-secretors of this molecule, making it ineffective in these individuals. Finally, CA 19-9 testing is not a perfect indicator of disease progression or response to treatment. While decreases in CA 19-9 levels may indicate a positive response to treatment, this marker is not always reliable in predicting the success of therapy or the likelihood of recurrence [6]. One must consider the clinical context, patient characteristics, and other diagnostic tests when interpreting results from CA 19-9 testing.

Other alarm signs/symptoms that should be kept in mind are:Diagnosis of hereditary/tropical pancreatitis;Reappearance of pain after pain relief;Appearance of obstructive jaundice;Markedly dilated pancreatic duct on imaging;Unexplained weight loss despite pancreatic enzyme replacement therapy;Pancreatic head mass on imaging;Vascular invasion on imaging.

Another red-flag sign is newly developed diabetes or recently deteriorated previously well-controlled diabetes [7]. In CP with jaundice, a palpable gallbladder is quite rare and raises the possibility of malignancy [8]. Due to the total and progressive nature of obstruction in malignant biliary stricture, the jaundice is usually deep in the background of CP. Standard imaging methods, including CT, MRI, and ultrasound abdomen, offer additional data and aid in separating these entities. Unfortunately, it can occasionally be challenging to distinguish between an inflammatory tumour and malignancy, due to extensive overlap of clinical, biochemical, and imaging characteristics. This is corroborated by the discovery of an inflammatory mass masquerading as pancreatic carcinoma in a prior large study of pancreatic resections for carcinoma head of pancreas, which was seen in 6.5% of cases [8].

## 5. Evaluation of Suspicious Malignancy in CP

### 5.1. Imaging Modalities

**Abdominal ultrasound:** Ultrasound is a relatively inexpensive, widely available, and non-invasive imaging modality without the risk of contrast-associated adverse events. It is usually the first-line investigation in the evaluation of a patient presenting with obstructive jaundice. However, the real-world accuracy of abdominal ultrasound in the diagnosis of pancreatic tumours ranges from 50–70% [9]. It is highly operator-dependent. The limiting factors are obesity, increased bowel gas, and patient discomfort, all of which limit the use of this modality in the evaluation of the pancreas. Therefore, if, in a patient presenting with obstructive jaundice, an initial ultrasound rules out choledocholithiasis or biliary tumour and a pancreatic aetiology is suspected, a contrast-enhanced computed tomography (CECT) or MRI abdomen would be the next logical step.

**Contrast-Enhanced CT scan:** Use of CT for diagnosing abdominal lesions has the advantages of reduced acquisition time with coverage of large volumes, allowing multiphase data acquisition in submillimetre slices. CT with IV contrast allows better visualisation of the tumour with respect to vascular structures, which can help in both early detection and staging of the disease. It has a reported sensitivity between 76–92% for diagnosing pancreatic adenocarcinoma [10,11,12,13,14]. A dedicated pancreatic protocol CT has reported better sensitivity and specificity in the diagnosis of pancreatic masses. This method uses triphasic imaging protocols comprising an arterial phase (a delay of 15–30 s), a pancreatic phase (a delay of 45–50 s), and a venous phase (a delay of 70 s). The rationale behind this imaging protocol is the improved pancreas-to-lesion contrast during pancreatic phase imaging, whereas a tumour may be missed on single-phase imaging. Based on the original thin-slice dataset, secondary three-dimensional reconstruction techniques may be performed including maximum intensity projection (MIP), volume rendering technique or surface shaded display mode; thus, different multiplanar views of the region of interest and adjacent vascular structures may be obtained.

**Magnetic Resonance Imaging (MRI):** Patients with equivocal ultrasound and/or CT results with high indices of suspicion for malignancy can benefit from an abdominal MRI. The most widely used agent is gadolinium, which is usually administered intravenously (IV). Because of the lesion’s hypovascularity and extensive fibrous stroma, PDAC looks hypo-intense on T1 weighted contrast sequences. On delayed images, tumours look isointense as a result of the contrast medium’s sluggish wash-in. Contrast-enhanced CT has a sensitivity of 86%, compared to conventional MRI’s 84%, hence conventional MRI does not have any additional significant advantage [14]. There is no added benefit from combining the two imaging modalities. MRI is more effective at defining pancreatic cystic lesions and can help find a malignant focus within such lesions.

**Magnetic Resonance Cholangiopancreatography (MRCP):** MRCP uses either thick slab or thin slab turbo spin echo T2 sequences (TSE), which allow dedicated ductal visualisation. It is better than CT in evaluating both pancreatic and biliary ductal anatomy [14]. It can visualize the ductal anatomy both above and below a stricture and can also detect any intrahepatic mass lesions. In differentiating chronic pancreatitis from pancreatic carcinoma, detection of additional secondary signs such as the “duct-penetrating sign” is helpful.

### 5.2. Imaging Features to Differentiate CP from PDAC

PDAC and inflammatory mass in CP are generally hypo-attenuating and hypo-enhancing. An iso- or hypo-intense mass at T1-weighted MRI and an iso- or hyper-intense mass at T2-weighted MRI, which are findings that are comparable to those of PDAC, are the main imaging hallmarks of mass-forming pancreatitis. Fat-suppressed sequences can increase diagnostic confidence. The diagnosis of an inflammatory head mass should be more likely if parenchymal calcifications or pseudocysts are present [10,11,12,13,14,15]. The distinction between a malignant and an inflammatory head mass can be made with the aid of specific secondary imaging findings. A number of auxiliary imaging signs aid in separating CP and PDAC (Table 1).

### 5.3. Endoscopic Modalities

A significant advancement in the evaluation of pancreatic disease, particularly lesions of the pancreatic head, has been the introduction of endoscopic ultrasound (EUS). Additionally, with the advent of novel EUS-based imaging techniques such as digital image analysis, EUS elastography, and contrast-enhanced EUS, it is now possible to characterise pancreatic lesions more accurately, especially when they are occurring against a background of chronic inflammation.

**Conventional EUS:** The pancreas can be precisely targeted with EUS, reducing air and bone interference and enabling higher frequency, higher resolution images. Moreover, curved linear array echoendoscopes have the added benefit of FNA/FNB, which helps with histological confirmation. Furthermore, intraductal imaging can be provided by passing EUS catheter probes through the ampulla. When other methods have failed to diagnose pancreatico-biliary diseases, EUS with or without FNA has been demonstrated to be a cost-effective alternative [16]. Regardless of pathology, the majority of solid pancreatic lesions appear as heterogenous hypo-echoic masses. The reported median sensitivity of EUS for pancreatic tumour identification is 94% [17,18,19]. In a recent study of 120 patients, the sensitivity of EUS was higher than that of computed tomography (98% vs. 74%) [17]. Moreover, trans-abdominal ultrasound was found to have a lower sensitivity (94% vs. 67%) than EUS [17]. However, comparative studies between MRI and EUS are scarce. EUS is very helpful for the detection of small pancreatic lesions, due to its excellent spatial resolution. The sensitivities of EUS, CT, and MRI were 93%, 53%, and 67%, respectively, in a study assessing the effectiveness of various modalities for detecting pancreatic tumours under 30 mm in diameter (*n* = 49) [20]. The accuracy in distinguishing benign inflammatory from malignant tumours is no higher than 75% with EUS, despite its high-resolution capabilities [21,22,23,24,25]. This is partly because several characteristics of CP, such as peripheral calcifications, are also present in malignant tumours. Moreover, pancreatic cancer T-staging is challenging due to the inflammation seen in CP. These restrictions can be circumvented via EUS guided sampling.

The role of EUS–FNA in the diagnosis of solid pancreatic lesions has been reviewed extensively (Figure 1 and Figure 2). The reported sensitivity and accuracy for malignancy range from 75 to 92 percent and 79 to 92 percent, respectively [26,27]. It is questionable whether or not on-site pathologists should always be used to diagnose solid pancreatic lesions, given the financial and logistical burdens. However, EUS–FNA’s high sensitivity and diagnostic accuracy do not extend to mass lesions in the presence of chronic pancreatitis. In a series of 300 EUS–FNA procedures for pancreatic mass lesions, Varadarujulu et al. showed that sensitivity decreased to 73.9% in patients with chronic pancreatitis, as compared to 91.3% in those with a normal pancreas [28]. In a study by Iordasche et al., only 50% of the 72 patients with chronic pancreatitis—of whom 17 had pancreatic cancer—could be detected by EUS–FNA [29]. In a separate German study of thirteen patients with chronic pancreatitis and pancreatic carcinoma, EUS–FNA was only able to identify the cancer in seven of the patients [23]. In certain circumstances, increasing the number of passes or repeating the process may increase yield [30]. Moreover, with the introduction of newer-generation FNB needles, tissue yield has improved, with an anticipated higher accuracy [31]. An earlier comprehensive review and meta-analysis of nine randomised studies indicated that EUS–FNB had a better odds ratio (OR) than EUS–FNA, with a 95% confidence interval (CI) of 1.87 to 2.63 [31]. However, no comparative studies between FNA and FNB needles are available from these cohorts of patients. The utility of rapid on-site evaluation (ROSE) of a biopsy specimen by an experienced pathologist in this setting also has not been studied.

Under challenging circumstances, using molecular techniques can aid in increasing yield. The efficacy of microsatellite markers and K-ras gene mutations in EUS–FNA samples from patients with benign lesions and pancreatic tumours was demonstrated by a study by Khalid et al. [32]. The diagnostic yield of FNA was increased by the introduction of mutational markers, since the mean fractional mutation rate was higher in malignant tumours. K-ras mutations and allelic loss in tumour suppressor genes were identified on EUS–FNA specimens in a study of a similar nature involving 101 participants [33]. The K-ras gene mutation, p16 allelic loss, and DPC4 gene mutation all increased the sensitivity of cancer detection to 100%. Using K-ras mutation testing increased diagnostic sensitivity for malignancy to 88%, which was only slightly better than cytopathology alone (83%), according to a different and sizable prospective multicentre study [34]. The absence of the K-ras mutation, however, was a very reliable indicator that the lesion was benign. This emphasises the value of exploring several markers rather than just one. The absence of K-ras mutation in FNA samples from patients with chronic pancreatitis and a mass lesion strongly implies a benign aetiology, according to other studies of a similar kind [35]. To summarise, the aforementioned studies show that molecular tests are crucial for identifying pancreatic cancer in FNA samples, and that assessing for K-ras mutations and the loss of tumour suppressor genes will help to increase accuracy.

**Contrast enhanced EUS (CEUS):** Another method for improving EUS-based diagnosis of solid pancreatic lesions is to administer contrast agents. After injection, the arterial phase lasts about 30 s and the venous phase lasts about 90 s (Figure 3). Initial research indicated that injection of contrast agents could be useful in detecting malignant vascular infiltration by demonstrating the hyper-vascular nature of neuroendocrine tumours and the hypo-vascular nature of pancreatic adenocarcinoma [36,37,38,39]. The current technique employs a dedicated contrast-harmonic echo sequence to detect signals from micro bubbles delivered by newer second-generation contrast agents such as Sonovue and Sonazoid. Fusaroli et al. demonstrated that the presence of a hypo-enhancing mass with an inhomogeneous pattern confirms the diagnosis of pancreatic adenocarcinoma with 96% sensitivity and 86% accuracy [40]. This study also found that this pattern detected malignancy more accurately than simply finding a hypoechoic mass on a conventional EUS. The role of quantitative CE-EUS in the diagnosis of pancreatic cancer and chronic pancreatitis was investigated by Seicean et al. The most frequent finding was a hypo-enhanced pattern in both mass-forming chronic pancreatitis (10/12 patients) and pancreatic cancer (14/15 cases). When a contrast-uptake ratio index was established, it was discovered that cases of adenocarcinoma had much lower values than did cases of mass-forming chronic pancreatitis. A cut-off uptake ratio index value of 0.17 was determined for the diagnosis of adenocarcinoma, yielding an AUC of 0.86 (95% CI: 0.67–1.00), with a sensitivity of 80%, specificity of 91.7%, positive predictive value of 92.8%, and negative predictive value of 78% [41]. Differences in histology, such as the degree of fibrosis and the number of blood vessels obliterated in the tumour, may be related to variations in enhancement behaviour.

Furthermore, the use of pulsed Doppler can aid in the differentiation of adenocarcinoma from mass-forming CP. Pancreatic adenocarcinomas exhibit predominantly arterial-type signals, whereas chronic pseudotumoral masses exhibit both arterial and venous-type signals [42].

**EUS elastography:** The first study using EUS elastography to assess pancreatic tissue was published in 2006. Registering the variations in distortion of the EUS image following the application of mild pressure by the EUS probe, it is a non-invasive approach that assesses tissue elasticity (Figure 4). The elasticity of the tissue can be altered by a variety of pathological conditions, including inflammation, fibrosis, and malignancy, all of which will provide a distinctive elastographic appearance. The earliest investigations on elastography that were published relied on qualitative assessment with a colour-scale depicting various levels of tissue elasticity. Giovanni et al. used a scoring system based on various colour patterns in EUS elastography pictures to analyse 24 pancreatic masses. It was found to have a 100% sensitivity and 67% specificity [43]. The sensitivity and specificity of EUS elastography to distinguish benign from malignant pancreatic lesions were 92.3% and 80.0%, respectively, in a multicentre trial with 121 pancreatic masses, as opposed to 92.3 and 68.9%, respectively, for the conventional B-mode imaging [44]. The interobserver agreement was strong (kappa score = 0.785). Iglesias-Garcia et al. examined 130 patients with solid pancreatic masses in a similar study. With sensitivity, specificity, positive and negative predictive values, and overall accuracy of 100%, 85.5%, 90.7%, 100%, and 94.0%, respectively, qualitative EUS-elastography could be utilized to diagnose malignancy [45]. However, the use of qualitative elastography in the setting of chronic pancreatitis is questionable. Hirche et al. reported their findings of 70 patients with unclassified solid pancreatic lesions, determining that only 56% of patients could receive an adequate evaluation, and that EUS elastography had poor diagnostic sensitivity (41%), specificity (53%) and accuracy (45%) in predicting the nature of pancreatic lesions [46]. Software can quantify tissue strain to offer strain ratios that are different in benign and malignant lesions, overcoming the limitation of subjective error with qualitative elastography. Using quantitative elastography, a numerical result is produced, either as the average colour in a chosen area (mean hue histogram) or as the target area’s elasticity relative to soft reference tissue (strain ratio). In a study by Iglesias-Garcia et al., using quantitative elastography, eighty-six patients with solid pancreatic masses (forty-nine adenocarcinomas, twenty-seven inflammatory masses, six malignant neuroendocrine tumours, two metastatic oat cell lung cancers, one pancreatic lymphoma, and one pancreatic solid pseudopapillary tumour) and twenty controls were included. The mean strain ratio in healthy pancreatic tissue was 1.68 (95% CI: 1.59–1.78). The strain ratio for inflammatory masses was significantly greater than that of the healthy pancreas (mean 3.28; 95% CI: 2.61–3.96; *p* = 0.001) but significantly lower than that of pancreatic cancer (mean 18.12; 95% CI: 16.03–20.21; *p* = 0.001). The largest strain ratio was found in endocrine tumours (mean 52.34; 95% CI: 33.96–70.71). With a cut-off value of 6.04, the strain ratio’s sensitivity and specificity for diagnosing pancreatic cancers were 100% and 92.9%, respectively, surpassing the precision attained with qualitative elastography [47].

The contrast enhancement and elastography techniques can also be employed in tandem. In a study that combined the aforementioned approaches, the positive predictive value for diagnosing pancreatic cancer and chronic-pancreatitis-related pseudotumors was 96.7% [48]. The outcomes of these new procedures are encouraging, but the equipment employed has an impact. It will be simpler to integrate these methods into clinical practise if standard guidelines are developed and they are followed consistently.

### 5.4. Other Modalities

#### Perfusion Weighted MRI

A recently developed imaging technique called high field magnetic resonance perfusion imaging, also known as perfusion-weighted MRI, is used to examine the intralesional hemodynamics in pancreatic cancer cases. Dynamic contrast-enhanced MRI is the method that is most frequently utilised. As a result of this method, certain quantitative parameters may give new hints about focal pancreatic lesions. A recent study has shown that PDAC and chronic pancreatitis have different time–signal–intensity curves. While focal mass in a case of CP exhibits rapid enhancement followed by a signal plateau, PDAC has a type 2 time-signal intensity curve with contrast augmentation followed by slow progressive enhancement [49].

### 5.5. Intraoperative Evaluation

When there is a high clinical likelihood of malignancy and the diagnosis is still in question despite a thorough imaging assessment, pancreaticoduodenectomy is the recommended surgical procedure. In suspicious situations, a diagnostic laparoscopy may detect metastatic disease that was initially overlooked on preoperative screening. Although intraoperative ultrasonography (IOUS) is frequently employed, it might be challenging to tell an inflammatory head mass from a malignant tumour. In over 50% of cases, the routine use of IOUS for pancreatic surgery was reported to be helpful, and it was reported beneficial in 22.9% of patients [50]. Malignancy is confirmed by obvious features such as superior mesenteric vein, portal vein, or splenic vein, or superior mesenteric artery involvement. A study found that IOUS was very accurate in assessing suspicious cases of PDAC, and that combining biopsy with histological evaluation did not improve its diagnostic value [51]. After numerous surgical biopsies, peritoneal tumour dissemination is still a cause of major concern. Moreover, negative or ambiguous biopsies worsen the predicament. Due to the extremely high rate of false negative results (70%) and correspondingly low sensitivity of frozen sections, earlier publications have questioned their usefulness [52]. By considering the major and minor criteria listed by Hyland et al. [53] in Table 2, a diagnosis can be made.

Core biopsy was reported to have higher accuracy than wedge biopsy (77–86 vs. 38–75%) [54,55]. While being performed under vision, wedge biopsy faces the risk of only collecting a superficial sample due to concern over damaging the pancreatic duct. Complete excavation of the pancreatic head in patients with suspected head masses can produce more pancreatic tissue for frozen sections, increasing diagnostic assurance in ruling out malignancy, as demonstrated by Fancellu et al. [56]. According to recent studies, the accuracy of frozen section and histological interpretation has grown from 65 to 75 percent in the 1980s to above 90% [57]. However, due to the presence of reactive duct alterations and atrophy, the false negative rate ranges from 1.2 to 30% and is relatively high in the context of chronic pancreatitis [58]. The assessment of frozen section samples for dedifferentiation markers is one way to improve diagnostic accuracy [59]. Protein expression of CD-97, CD-95, and Fas-L can discriminate between normal parenchyma or pancreatitis from PDAC. Almost 12% of these lesions go undetected despite a thorough assessment [59]. In the suspected intraductal locations, intraoperative endocytoscopy can be useful in verifying the diagnosis. In order to identify PDAC in CP, this approach involves local spraying of dyes such methylene blue, followed by evaluation with an endocytoscope to search for characteristic surface histology of the diseased portion. The effectiveness of this procedure is still up for debate because it goes against oncological principles to open the duct for endocytoscopy.

## 6. Prognosis

When matched stage for stage, the results of a curative pancreaticoduodenectomy for a resectable PDAC with or without CP are comparable. Due to the tissue’s solid structure, patients with background CP are predicted to have a reduced postoperative fistula rate, albeit Chu et al.’s study found comparable rates [60]. Splanchnic venous thrombosis presenting concurrently may complicate dissection and cause intraoperative haemorrhage. In cases with a basis for the theoretical presumption of an increased cumulative risk of PDAC in patients with CP, total pancreatectomy can be considered as an option in a patient with PDAC and CP.

## 7. Special Subtypes

### 7.1. Paraduodenal or “Groove” Pancreatitis

PDP, also known as cystic duodenal dystrophy, is a kind of pancreatitis that centres on the pancreaticoduodenal groove. It may also be a site of adenocarcinoma infiltration. Inflammation or fibrosis that results can mimic locally-invasive PDAC of the pancreatic head by forming a pseudotumor that may spread into the nearby pancreatic head. PDP has been divided into three separate subgroups, each with unique histopathological and imaging findings [61]. The PDP type 1 solid tumour, which appears as a solid pseudotumor with little cystic alteration (Figure 5), may appear as a solid expansile lesion involving the pancreatic head or as a sheet-like mass in the pancreaticoduodenal groove. The latter could be especially challenging to detect from PDAC. In contrast, type 2 PDP is much simpler to distinguish, since the pancreaticoduodenal groove has undergone cystic alteration. More than 80% of the lesions in this case are cysts. The type 3 PDP, which is ill-defined and not similar to a mass, is less likely to resemble a malignancy.

A solid or cystic mass localised at the pancreaticoduodenal groove is one of the key imaging characteristics that point to a diagnosis of PDP in the differential diagnosis. The caveat that 20% of individuals have no apparent cysts in the lesions may encourage investigation of PDP when there are cystic changes [49]. Even in the absence of obvious cysts, a sheet-like mass, sometimes known as a “sandwich sign”, with a linear mass centred in the groove, is strongly predictive of type 1A PDP. It is challenging to distinguish Type 1B PDP’s solid “rice-ball pattern” from PDAC. A non-neoplastic aetiology may be suggested by the absence of biliary dilatation or a lack of significant pancreatic parenchymal atrophy [62]. PDP may be diagnosed if there are other symptoms, such the duct penetrating sign, thickening of the medial duodenal wall, or enlargement of the space between the ampulla and duodenal lumen [63]. Common bile duct displacement and/or gastroduodenal artery encasement are more likely to indicate PDP than PDAC. Similar to PDAC, PDP exhibits a hypo-intense arterial phase on contrast-enhanced CT with gradual late phase enhancement from fibrosis. It may be easier to rule out a pancreatic origin for the lump if a fat plane can be visualised separating it from the organ. Distinguishing the difference from PDAC may be aided by the ability to distinguish involvement of nearby structures such as the common bile duct or nearby vessels such as the gastroduodenal artery. The pancreaticoduodenal groove mass in PDP can mimic PDAC on MRI because it is iso- or hypo-intense on T1 weighted sequence and iso- or hyper-intense on a T2 weighted sequence. The presence of microcysts in the mass on the T2 weighted sequence points to PDP as the cause of the mass. On MRCP, PDP may result in a smooth enlargement of the pancreatic duct and distal common bile duct, as well as a wider gap between the ampulla and the intestinal lumen. A “double-duct sign” could be caused by fibrosis in the ampulla’s vicinity [64].

A study by Muraki et al. found that during the resection of non-neoplastic lesions, PDP was found in approximately 27% of cases, and as many as two-thirds of cases had a pre-surgical diagnosis of pancreatic or prepapillary cancer [61]. Treatment options include medical or endoscopic management in the form of pancreatic sphincterotomy, stent placement, or cyst drainage. Although Whipple’s procedure has been shown to improve patient outcomes, it can be associated with significant morbidity; hence, making an accurate pre-operative diagnosis is imperative.

### 7.2. Autoimmune Pancreatitis

Clinicians frequently struggle with the diagnosis of autoimmune pancreatitis since it is a distinct clinical entity with a variety of clinical, imaging, and histological characteristics. In a case of chronic pancreatitis with high immunoglobulin levels, Yoshida et al. demonstrated a prompt response to steroids, establishing the theory of autoimmune pancreatitis [65]. Two types of AIP (Table 2) have been described primarily on histological grounds: type 1 lymphoplasmacytic sclerosing pancreatitis and type 2 idiopathic duct centric pancreatitis [66]. Type 1 has extra pancreatic diseases and is characterised by increased immunoglobulin levels. Steroid therapy usually improves symptoms. Type 2 AIP can be difficult to identify and distinguish from PDAC. In 85% of instances, a localised pancreatic mass that closely resembles cancer is present along with immunoglobulin 4 levels that may be normal [67]. Except for inflammatory bowel disease, which may be present in 30% of cases and may provide a hint to the diagnosis, extra-pancreatic solid organ involvement is uncommon. The HISORt criteria suggested by the ICDC guidelines are traditionally used to make diagnoses [68].

Clinically, AIP symptoms can closely resemble those of PDAC, especially if they develop along with steatorrhea, weight loss, and painless jaundice. Though it seldomly occurs, PDAC might exhibit the typical trait of elevated IgG4 levels in type 1 AIP. When separating type 1 AIP from pancreatic cancer, a greater threshold (i.e., >2 × ULN of IgG4 levels) boosts specificity to 99% while decreasing sensitivity [69]. It is harder to distinguish type 2 AIP from pancreatic cancer because, unlike type 1, it rarely manifests with elevated IgG4 levels. Both can exhibit a cholestatic pattern of elevated liver function.

According to the ICDC guidelines, patients who appear with obstructive jaundice can be diagnosed with AIP or pancreatic cancer with a high degree of certainty based on imaging results [68]. Classic imaging characteristics of the diffuse form of AIP type 1 include disappearance of lobulations, a smooth contour, and a diffuse sausage-like expansion of the gland. When present, a low-attenuation rim or capsule-like halo is pathognomonic for type 1 AIP. It is typical to see homogenous contrast enhancement in both the early and delayed periods [70]. Although reactive lymphadenopathy can occur, calcifications and pseudocysts are usually absent. The imaging appearance of focal AIP may closely resemble that of a tumour in the absence of diffuse illness. The mass may have ambiguous borders, and periglandular inflammation may resemble extraglandular malignancy spread [71]. The double-duct sign may also be caused by concurrent common bile duct involvement, which makes differentiating it from PDAC much more challenging. Multiple bile duct strictures and pancreatic duct strictures without side-branch dilatations may suggest a diagnosis of AIP in the absence of the pathognomonic capsule-like rim. The duct-penetrating sign, which AIP exhibits, further supports a benign diagnosis. The recognisable fibrotic rim or “halo” seen on CT scans is T2 hypo-intense on MRI and exhibits little contrast enhancement. Pancreatic duct strictures and the duct penetrating sign may be found with MRCP. Several pancreatic duct strictures and/or concurrent common bile duct involvement suggest an AIP diagnosis rather than a PDAC diagnosis. Further hints may be provided by incidental IBD or secondary sclerosing cholangitis findings. Diffusion-weighted MRI has a demonstrated value in the diagnosis of AIP and tracking the therapeutic response. AIP can be diagnosed with a sensitivity of 83% and differentiated from PDAC with a specificity of 79%, according to Hur et al. [70]. Unfortunately, this method is insufficient to distinguish AIP from PDAC and additional investigations may be warranted.

The presence of some additional signs can add to the diagnostic confidence for these patients. Previous studies have shown that the capsule-like rim sign has a specificity of 97–100% for separating AIP from PDAC, while sensitivity is only 29% [71]. Similar to skip strictures in the main pancreatic duct, skip strictures in the common bile duct have been demonstrated to have 100% specificity but low sensitivity at 33% and 44%, respectively [72]. In order to distinguish between inflammatory and malignant masses, early studies have shown the extra diagnostic utility of perfusion-weighted MRI employing dynamic contrast-enhanced sequence.

It is challenging to distinguish AIP from pancreatic cancer based on hypoechoic masses observed on traditional EUS. Conventional EUS revealed evidence of diffuse hypoechoic gland enlargement, thickening of the bile duct wall, and surrounding hypoechoic zones in AIP as compared to PDAC, according to Hoki et al. [73]. AIP is known to exhibit chronic pancreatitis-like signs, such as hyperechoic foci or strands. Okabe et al. revealed that these effects can remain despite steroid therapy [74]. The relevant caveat is that imaging findings are dependent on the stage of AIP at the time of diagnosis. Time intensity curves with CH-EUS were shown to be helpful for distinguishing AIP from pancreatic cancer in a study by Imazu et al. The peak intensity (PI) and maximum intensity gain (MIG) values were significantly higher in patients with AIP, as compared to pancreatic cancer [75]. Preoperatively, EUS elastography has been utilised to differentiate pancreatic masses. According to Mei et al.’s meta-analysis, the sensitivity, specificity, and odds ratio of elastography in distinguishing benign and malignant solid pancreatic lesions were 0.95 (95% CI: 0.94–0.97), 0.67 (95% CI: 0.61–0.73), and 42.28 (95% CI: 26.90–66.46), respectively [76]. Dietrich et al. discovered distinctive elastographic patterns not only at the site of AIP masses but also in the surrounding pancreatic tissue [77].

While ruling out pancreatic cancer is crucial in the evaluation of any pancreatic mass lesion, obtaining pancreatic tissue is critical in distinguishing AIP from pancreatic cancer. With samples collected with EUS–FNA, pancreatic cancer can be diagnosed histopathologically with a very high degree of accuracy. An exceptionally high diagnostic ability was found by Chen et al. in a meta-analysis of the histopathological diagnostic capacity of EUS–FNA confined to pancreatic cancer, with a pooled sensitivity of 0.89 (95% CI: 0.88–0.90) and a pooled specificity of 0.96 (95% CI: 0.95–0.97) [78]. The relevance of AIP histological diagnosis is also supported by the international consensus diagnostic criteria for autoimmune pancreatitis [62]. A recent randomized controlled trial by Kurita et al. to compare the diagnostic yields of 22G Franseen tip vs. forward bevel tip FNB needles shows superior yields with the Franseen tip (78% vs. 45%) and advocates its routine use in the diagnosis of AIP [79]. In the era of precision medicine, histopathological diagnosis of AIP and its need to differentiate from pancreatic malignancy will play an increasingly important role in patient management.

## 8. Prevention and Screening for Pancreatic Cancer in Background of CP

There are no established recommendations for PDAC screening yet. United States preventive services task forces have recommended against screening of the general population for PDAC. Currently, routine pancreatic cancer screening is advised for individuals who have high-risk conditions such Peutz–Jeghers syndrome or a family history of the disease. In those circumstances, it may be possible to perform either a highly sensitive serological test or non-invasive imaging. The diagnostic usefulness of CA-19-9, which has a low sensitivity of 80% and a specificity of 75%, is still limited [80]. The NCCN recommendations advise using EUS in these high-risk patients [81]. Among 216 asymptomatic high-risk participants in a multicentre prospective cohort study from the United States, pancreatic abnormalities were found by CT, MRI, and EUS in 11.0%, 33.3%, and 42.6% of cases, respectively [82]. EUS, despite being very sensitive, is not without its limitations as a screening test due to high numbers of false positive and false negative results, especially in the background of CP. Furthermore, whether this technique improves results and the optimal frequency of screening exams are both debatable [83]. The use of molecular markers for screening in biopsy or FNA samples, such as miR-16 and mutational KRAS analysis, and expression analysis of UHRF1, ATP7A, and aldehyde oxidase 1 is still being studied [84]. Preventive measures appear logical given the poor results of PDAC. Nevertheless, there are no effective preventative measures in place. A multicentre study from Japan found that patients who had their CP surgically treated—the majority of whom received combined drainage and resection—had a lower incidence of PDAC (0.7 vs. 5.1%, *p* = 0.03; hazard ratio: 0.11) [85]. While the follow-up in this trial was only for five years, longer-term investigations may produce more reliable results.

## 9. Conclusions

Diagnosis of pancreatic cancer in the background of chronic pancreatitis requires high clinical suspicion and judicious use of investigations (Figure 6). Standard cross-sectional imaging such as CT/MRI and functional imaging can help in differentiating malignancy from an inflammatory head mass; however, they both have poor negative predictive value. With the advent of newer technologies such as EUS—FNB needles, CE-EUS, and EUS elastography, there is an unmet need for development of standardized protocols, consensus, and operator training before these techniques can be used routinely.

## Figures and Tables

**Figure 1 diagnostics-13-01797-f001:**
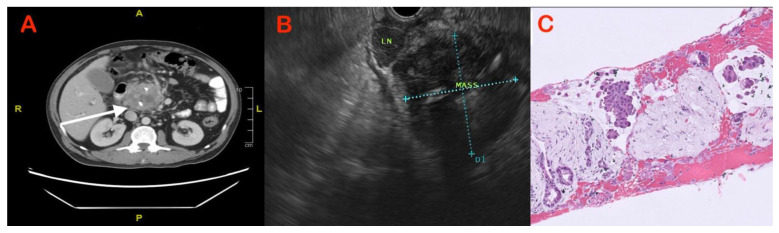
Case of chronic pancreatitis with mass in head of pancreas: (**A**) CT showing displaced calcifications (arrow) with hypodense mass; (**B**) heterogeneously hypoechoic mass seen on EUS with perilesional lymph nodes; and (**C**) adenocarcinoma seen on histopathology sections obtained via EUS–FNB.

**Figure 2 diagnostics-13-01797-f002:**
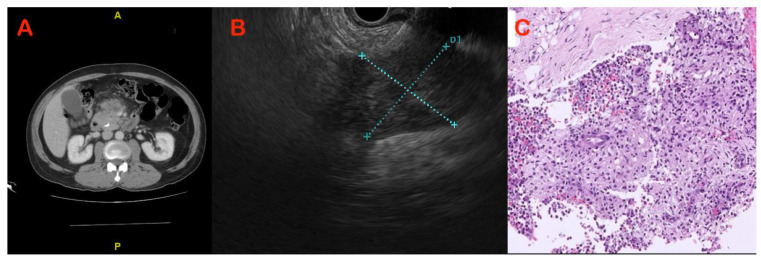
Case of chronic pancreatitis with mass in neck of pancreas: (**A**) CT showing hypodense mass with perilesional stranding; (**B**) hypodense mass in close proximity to mesenteric vessels on EUS; and (**C**) EUS–FNB showing dense granulomatous inflammation suggestive of tuberculosis.

**Figure 3 diagnostics-13-01797-f003:**
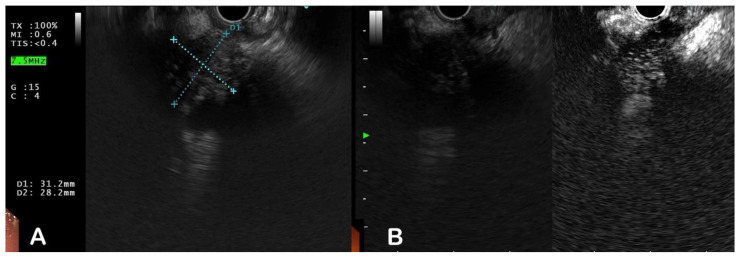
(**A**) Heterogenously hypoechoic mass lesion in head of pancreas on EUS; (**B**) CH-EUS showing an isoenhancing pattern suggestive of a possible inflammatory mass lesion.

**Figure 4 diagnostics-13-01797-f004:**
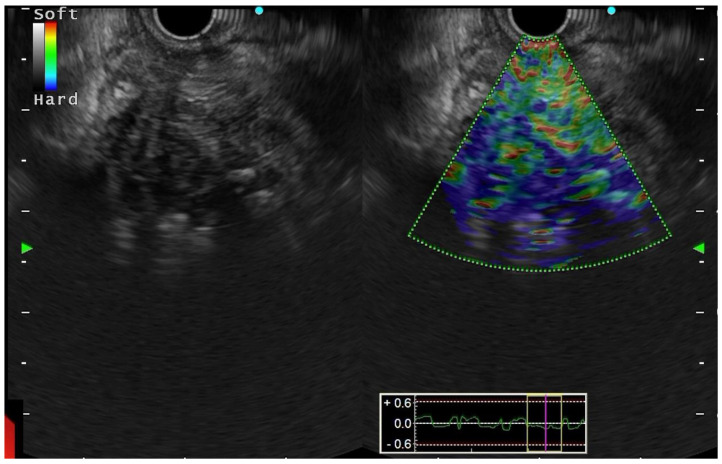
EUS Elastography of mass lesion in head of pancreas in background of chronic pancreatitis. Qualitative elastography using colour plots shows a diffusely variable stiffness in the mass likely suggestive of an inflammatory mass lesion. The strain ratio was calculated to be 4.1, and FNB confirmed the inflammatory mass.

**Figure 5 diagnostics-13-01797-f005:**
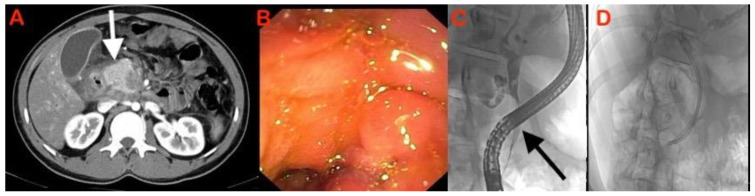
(**A**) CT showing hypodense mass lesion in pancreaticoduodenal groove which is enhancing on arterial phase—suggestive of groove pancreatitis (arrow); (**B**) duodenal infiltration seen on upper endoscopy; and (**C**) ERCP showing stricture in distal CBD (arrow). (**D**) Multiple plastic stents were placed. The patient showed improvement with conservative management over next 3 months.

**Figure 6 diagnostics-13-01797-f006:**
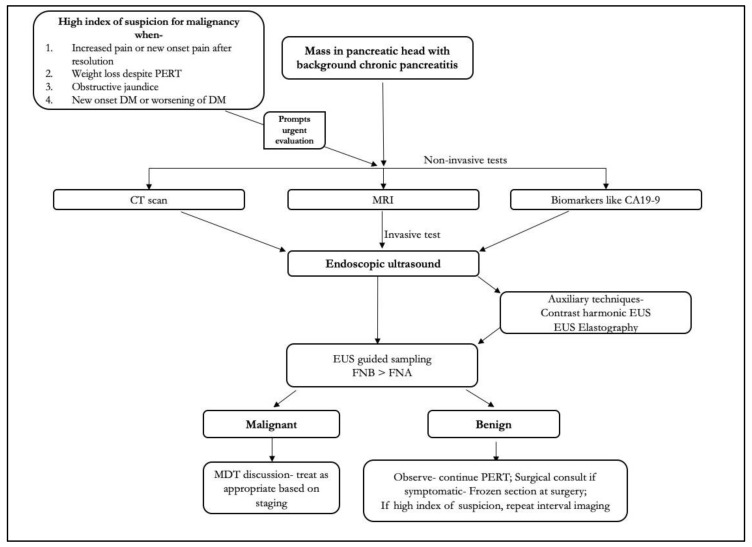
Algorithm for approach to mass in pancreas in background of chronic pancreatitis.

**Table 1 diagnostics-13-01797-t001:** Auxiliary imaging features to differentiate CP from PDAC.

Auxiliary Imaging Features to Differentiate CP from PDAC	
**1.** **Duct penetrating sign**	Smooth narrowing of the pancreatic duct as it traverses through the mass without any abrupt cut-off is a reliable sign that it is inflammatory. The diagnostic accuracy of this sign is 94%.
**2.** **Side branch dilatation**	Presence of side-branch dilatation is a reliable sign that the mass is inflammatory in nature. This phenomenon is hypothesised to occur due to the traction effect caused by interstitial fibrosis in chronic pancreatitis, rather than mass effect from a neoplasm where duct obliteration would be expected.
**3.** **Duct to parenchyma ratio**	PDAC is characterised by marked ductal dilatation and parenchymal atrophy. On EUS, a ratio of the diameters of MPD to parenchyma greater than 0.34 strongly suggests malignancy.
**4.** **Displaced calcifications**	In patients with underlying CP who develop a malignancy, the mass displaces the calcifications to the periphery.
**5.** **Double duct sign**	Simultaneous dilatation of both pancreatic and common bile ducts is an indicator of malignancy. It is seen in ampullary tumours and in 77% of the cases of pancreatic head malignancy; however, it is not exclusive to this, as it may also be seen in mass forming AIP as well as in other non-malignant conditions.
**6.** **Vessel encasement and deformity**	Soft tissue encasement is a characteristic sign of extra glandular spread of PDAC. The SMV teardrop sign showing malformation of SMV to a shape resembling a teardrop may suggest SMV encasement. Circumferential narrowing and vessel deformity may also be seen.
**7.** **SMA to SMV ratio**	Enlargement of SMA relative to SMV with an SMA to SMV ratio greater than 1.0 is a sign favouring the diagnosis of malignancy. Release of vasoactive substances in acute pancreatitis results in an increase in diameter of the much more distensible SMV in comparison to SMA. In PDAC, the proposed hypothesis for dilatation of SMA is due to the increased resistance to blood flow or due to vessel wall infiltration.

**Table 2 diagnostics-13-01797-t002:** Diagnostic criteria for pancreatic masses on a frozen section, as proposed by Hyland et al.

Major Criteria	Minor Criteria
Nuclear size variation of 4:1 or greater between ductal epithelial cells	Huge irregular epithelial nucleoli
Incomplete ductal lumen and disorganised duct distribution	Necrotic glandular debris
	Glandular mitoses
	Glands unaccompanied by connective tissue stroma within smooth muscle bundles
	Perineural invasion

## Data Availability

Not applicable.

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
