# Peer review of "Approach to Pancreatic Head Mass in the Background of Chronic Pancreatitis"

_diagnostics, 2023, doi:10.3390/diagnostics13101797_

Round 1

Reviewer 1 Report

I am pleased to have the opportunity to review this well written and interesting review. I have only a couple of comments.

First, one caution to raise is how to use the CA 19-9. This marker can be impact by a number of related but noncancerous findings such as biliary obstruction. But also there are a high percentage of so called CA 19-9 nonproducers. The authors indicate using CA 19-9 testing as a component of diagnosing the cancer, and it is in their figure as well. This should be at most in conjunction with other imaging modalities. I suggest that they comment on the limit of CA 19-9 testing and emphasize its use in conjunction with other tests.

Second, noting their introduction, is it correct that jaundice is typically associated with metastatic disease only? I consider locally advanced M0 cancers to be oftentimes identified by the painless jaundice.

There are a few instances where some proofreading will be useful: double periods and extra spaces and the like. Probably could run it through a grammar check software if need be.

Reviewer 2 Report

Introduction:

First paragraph: The only reference for the information cited is from 1985, which does not reflect current data. Please use current references.

Second paragraph: There are no references to the text. Please add relevant references. Please update the references in the first few sections of the manuscript to accurately reflect the cited material.

Section 4.1: Why is abdominal ultrasound abbreviated as USG? Please spell out the abbreviation the first time you write it (for example, in the same section where abdominal ultrasound is mentioned, you mentioned CECT for the first time).

4.2: Very good section, especially the table. Conventional EUS: what is ROSE in the last sentence?

Figure one: Please identify the calcification in A and the adenocarcinoma in C using arrows or arrowheads.

Figure two: Isn't the white spot also a calcification?

6.1: Need a reference to the end of the second paragraph.

Figure five: Please add identification to the images to make it clear to understand.

I would suggest removing figure 6 from the conclusion to a different section. If it is your own experience, you can explicitly mention and add a separate section; otherwise, place it in the respective section with appropriate reference.

.

I would recommend an English edit by a native speaker or editing program to correct some language mistakes.
